# Incidentally diagnosed cancer and commonly preceding clinical scenarios: a cross-sectional descriptive analysis of English audit data

Minjoung Monica Koo,[1] Greg Rubin,[2] Sean McPhail,[1,3] Georgios Lyratzopoulos[1]

[1]Epidemiology of Cancer Healthcare & Outcomes (ECHO) Group, Department of Behavioural Science and Health, University College of London, London, UK
[2]Institute of Health and Society, Newcastle University, Royal Victoria Infirmary, Newcastle upon Tyne, UK
[3]National Cancer Registration and Analysis Service, Public Health England, London, UK

**Correspondence to**
Dr Minjoung Monica Koo;
monica.koo@ucl.ac.uk

## ABSTRACT

**Objectives** Cancer can be diagnosed in the absence of tumour-related symptoms, but little is known about the frequency and circumstances preceding such diagnoses which occur outside participation in screening programmes. We aimed to examine incidentally diagnosed cancer among a cohort of cancer patients diagnosed in England.

**Design** Cross-sectional study of national primary care audit data on an incident cancer patient population.

**Setting** We analysed free-text information on the presenting features of cancer patients aged 15 or older included in the English National Audit of Cancer Diagnosis in Primary Care (2009–2010). Patients with screen-detected cancers or prostate cancer were excluded. We examined the odds of incidental cancer diagnosis by patient characteristics and cancer site using logistic regression, and described clinical scenarios leading to incidental diagnosis.

**Results** Among the studied cancer patient population (n=13 810), 520 (4%) patients were diagnosed incidentally. The odds of incidental cancer diagnosis increased with age (p<0.001), with no difference between men and women after adjustment. Incidental diagnosis was most common among patients with leukaemia (23%), renal (13%) and thyroid cancer (12%), and least common among patients with brain (0.9%), oesophageal (0.5%) and cervical cancer (no cases diagnosed incidentally). Variation in odds of incidental diagnosis by cancer site remained after adjusting for age group and sex.

There was a range of clinical scenarios preceding incidental diagnoses in primary or secondary care. These included the monitoring or management of pre-existing conditions, routine testing before or after elective surgery, and the investigation of unrelated acute or new conditions.

**Conclusions** One in 25 patients with cancer in our population-based cohort were diagnosed incidentally, through different mechanisms across primary and secondary care settings. The epidemiological, clinical, psychological and economic implications of this phenomenon merit further investigation.

## INTRODUCTION

Cancer is most often diagnosed following presentation with symptoms likely caused by the malignancy.[1 2] However, some patients

## Strengths and limitations of this study

► The findings are from a unique and large population-based cohort of patients diagnosed with a range of cancers with detailed characterisation of their presenting features.
► Diagnostic status (incidental or non-incidental) was identified using free-text information provided by healthcare professionals based on primary care records.
► We describe common mechanisms across different settings, and involving a range of tests and imaging modalities, that led to an incidental diagnosis of cancer.
► We were unable to compare clinical outcomes between incidentally and non-incidentally diagnosed cancer patients.

are diagnosed with cancer incidentally, in the absence of symptoms that could plausibly be related to the tumour and outside of formal cancer screening or surveillance activities. The use of imaging technologies (including X-ray, CT, MRI and PET scans) is a commonly described route to incidental diagnosis of different diseases, including cancer.[3–6] Chronic disease management involving periodic routine blood or urine testing are increasingly used in primary care and may represent another common pathway to incidental diagnosis.[7–10] Nonetheless, evidence regarding the frequency of such incidental diagnoses is currently limited.

Since incidental cancer diagnoses are characterised by the absence of tumour related symptoms, it is plausible that this may represent overdiagnosis in some patients, whereby the detected cancer would not have otherwise caused symptoms in the patient's lifetime.[11] Concerns about overdiagnosis thus far have largely focused on screening-detected cancers (eg, breast cancer), but it may be also occurring in other contexts.[12 13] Ahead of considering the clinical, psychological or

economic consequences associated with incidental diagnosis (including the potential for overdiagnosis), we need to address gaps in knowledge about the frequency and characteristics of incidentally diagnosed cancer.

We therefore aimed to examine the frequency of incidental diagnosis among an incident cohort of patients with cancer; compare the characteristics of incidentally versus non-incidentally diagnosed patients; and examine common pathways and mechanisms that led to the incidental diagnosis of cancer using a unique data source relating to a national quality improvement initiative in England.

## METHODS
### Study design and population
We analysed cross-sectional data collected as part of the English National Audit of Cancer Diagnosis in Primary Care (NACDPC).[14] Briefly, health professionals from 1170 participating general practices (representing 14% of practices in England) provided information on the diagnostic pathway for a consecutive sample of patients diagnosed with cancer during April 2009–2010. Participating practices were comparable to non-participating practices in (former) respective Cancer Networks, and the patient population was broadly representative of the contemporary national incident cancer patient cohort.[14 15] Unique to this audit, participating clinicians and other healthcare professionals provided information regarding the main presenting symptoms, cancer diagnosis, demographic characteristics and route of diagnosis for each patient based on primary care records.

### Definition and identification of cases
The nature of cancer diagnosis (incidental or non-incidental), was ascertained by examination of the free-text information included in the presenting symptoms data field (answering the audit question, 'What were the main presenting symptom(s) [of the patient]?').

Tumours were deemed to have been diagnosed incidentally if the incidental nature of diagnosis was explicitly recorded by the participating healthcare professional (indicated by phrases including 'accidental finding', 'chance finding', 'incidental', 'opportunistic'), or if the clinical circumstances described were consistent with incidental identification based on clinical knowledge (GL and GPR) and prior literature.[5 16 17] Cases were initially identified by MMK, and subsequently reviewed by GL and GPR; any disagreements were resolved by discussion.

Information was available on the patient's sex, age group, and cancer site (categorised as bladder, brain, cervical, colorectal, endometrial, gallbladder, leukaemia (of any type), laryngeal, liver, lung, lymphoma, melanoma, mesothelioma, multiple myeloma, oesophageal, oropharyngeal, ovarian, pancreatic, renal, sarcoma (of any type), small intestine, stomach, testicular, thyroid and vulval).[14] Patients diagnosed with prostate cancer were excluded a priori, given the difficulties in reliably distinguishing

reasons for prostate specific antigen testing.[18] Patients with screen-detected breast, colorectal and cervical cancer, and those diagnosed following surveillance for pre-malignant or high-risk conditions were also excluded. Therefore, the study population comprised 13 810 patients aged 15 or older with sufficient information to determine incidental/non-incidental status, and complete information on cancer diagnosis, age group and sex (see online supplementary figure 1 for sample derivation).

### Data analysis
First, we compared the demographic and clinical characteristics of incidentally and non-incidentally diagnosed patients. Logistic regression was used to calculate crude and adjusted ORs of incidental diagnosis by sex, age group and cancer site. Male patients, and those aged 60–69 years, were used as the reference category for sex and age group respectively, while colorectal cancer was used as the reference category for cancer site, as the most common non-sex specific cancer in our population. We also examined the cancer site case-mix ('cancer site signature') of the incidentally diagnosed group, that is, the relatively frequency of each cancer site among incidentally diagnosed patients. All statistical analyses were conducted in STATA SE V.15 (StataCorp).

Subsequently, we identified common clinical scenarios leading to incidental diagnosis based on a subgroup of patients with sufficient information (n=345, 66% of all incidental diagnoses). These findings were synthesised narratively.

### Sensitivity analysis
We performed sensitivity analyses expanding the definition of incidental diagnosis of cancer to include an additional 272 patients without any recorded presenting symptom, and/or with the presence of abnormal clinical findings indicated in response to the audit question, 'What were the main presenting symptom(s) [of the patient]?'

### Patient and public involvement
Patients and members of the public were not involved in the design of this study.

## RESULTS
### Incidentally diagnosed cancer patients
A total of 520/13 810 (4%) patients aged 15+ years were diagnosed incidentally with one of 25 cancer sites (other than prostate cancer). Men were more likely to be diagnosed incidentally than women (5% of men vs 3% of women), although there was no evidence to support this after adjustment for age and cancer site (see table 1). The odds of being diagnosed incidentally with cancer generally increased with age (joint Wald test p value <0.001).

Crude and adjusted ORs indicated substantial variation in the odds of incidental diagnosis between cancer sites (see figure 1 and table 1). Almost a quarter (23%)

**Table 1** Characteristics of incidentally vs non-incidentally diagnosed cancer patients, and crude/adjusted ORs of incidental status (n=13 810)

| | Total | Incidental | | Crude | Adjusted* |
|---|---|---|---|---|---|
| | N | n | % (95% CI) | OR (95% CI) | OR (95% CI) |
| Total | 13810 | 520 | 4 (3 to 4) | – | – |
| Sex | | | | 0.001† | 0.204† |
| Men | 5839 | 278 | 5 (4 to 5) | Ref. | Ref. |
| Women | 7971 | 242 | 3 (3 to 3) | 0.63 (0.53 to 0.75) | 0.88 (0.72 to 1.07) |
| Age group | | | | <0.001† | <0.001† |
| 15–49 years | 2072 | 31 | 1 (1 to 2) | 0.40 (0.27 to 0.59) | 0.39 (0.26 to 0.60) |
| 50–59 years | 2050 | 65 | 3 (2 to 4) | 0.86 (0.63 to 1.17) | 0.88 (0.64 to 1.21) |
| 60–69 years | 3181 | 117 | 4 (3 to 4) | Ref. | Ref. |
| 70–79 years | 3656 | 170 | 5 (4 to 5) | 1.28 (1.00 to 1.62) | 1.28 (1.00 to 1.64) |
| 80+ years | 2851 | 137 | 5 (4 to 6) | 1.32 (1.03 to 1.70) | 1.45 (1.12 to 1.89) |
| Cancer site | | | | <0.001† | <0.001† |
| Leukaemia‡ | 450 | 103 | 23 (19 to 27) | 10.49 (7.55 to 14.58) | 11.84 (8.49 to 16.51) |
| Renal | 356 | 46 | 13 (10 to 17) | 5.25 (3.53 to 7.78) | 5.60 (3.77 to 8.33) |
| Thyroid | 110 | 13 | 12 (7 to 19) | 4.74 (2.53 to 8.88) | 7.25 (3.80 to 13.82) |
| Liver | 103 | 11 | 11 (6 to 18) | 4.23 (2.16 to 8.27) | 4.42 (2.24 to 8.68) |
| Myeloma | 228 | 20 | 9 (6 to 13) | 3.40 (2.02 to 5.72) | 3.39 (2.01 to 5.70) |
| Gallbladder | 68 | 5 | 7 (3 to 16) | 2.81 (1.09 to 7.20) | 2.96 (1.15 to 7.62) |
| Mesothelioma | 75 | 4 | 5 (2 to 13) | 1.99 (0.71 to 5.61) | 1.88 (0.66 to 5.31) |
| Lymphoma | 698 | 33 | 5 (3 to 7) | 1.75 (1.14 to 2.69) | 2.10 (1.37 to 3.23) |
| Vulval | 73 | 3 | 4 (1 to 11) | 1.51 (0.46 to 4.94) | 1.70 (0.52 to 5.60) |
| Lung | 1875 | 77 | 4 (3 to 5) | 1.51 (1.08 to 2.12) | 1.49 (1.07 to 2.09) |
| Melanoma | 834 | 30 | 4 (3 to 5) | 1.32 (0.85 to 2.05) | 1.69 (1.09 to 2.64) |
| Bladder | 842 | 28 | 3 (2 to 5) | 1.22 (0.78 to 1.91) | 1.16 (0.74 to 1.82) |
| Colorectal | 2399 | 66 | 3 (2 to 3) | Ref. | Ref. |
| Stomach | 302 | 8 | 3 (1 to 5) | 0.96 (0.46 to 2.02) | 0.94 (0.45 to 1.99) |
| Ovarian | 394 | 10 | 3 (1 to 5) | 0.92 (0.47 to 1.81) | 1.11 (0.56 to 2.20) |
| Laryngeal | 121 | 3 | 2 (1 to 7) | 0.90 (0.28 to 2.90) | 0.96 (0.30 to 3.12) |
| Oropharyngeal | 213 | 5 | 2 (1 to 5) | 0.85 (0.34 to 2.13) | 1.05 (0.42 to 2.64) |
| Small Intestine | 53 | 1 | 2 (0.3 to 10) | 0.68 (0.09 to 4.99) | 0.72 (0.10 to 5.28) |
| Pancreatic | 370 | 6 | 2 (1 to 3) | 0.58 (0.25 to 1.35) | 0.59 (0.26 to 1.38) |
| Endometrial | 410 | 6 | 1 (1 to 3) | 0.52 (0.23 to 1.22) | 0.61 (0.26 to 1.43) |
| Testicular | 149 | 2 | 1 (0.4 to 5) | 0.48 (0.12 to 1.98) | 1.05 (0.25 to 4.43) |
| Breast | 2675 | 34 | 1 (1 to 2) | 0.46 (0.30 to 0.69) | 0.59 (0.38 to 0.91) |
| Sarcoma‡ | 106 | 1 | 0.9 (0.2 to 5) | 0.34 (0.05 to 2.45) | 0.43 (0.06 to 3.14) |
| Brain | 215 | 2 | 0.9 (0.3 to 3) | 0.33 (0.08 to 1.36) | 0.37 (0.09 to 1.54) |
| Oesophageal | 566 | 3 | 0.5 (0.2 to 2) | 0.19 (0.06 to 0.60) | 0.19 (0.06 to 0.60) |
| Cervical | 125 | 0 | 0 (0 to 3) | N/A | N/A |

*Adjusted for sex, age group and cancer site.
†Joint Wald test p value.
‡No information was available on leukaemia or sarcoma type.
N/A, not applicable.

of patients with leukaemia were diagnosed incidentally. More than one in ten patients with renal cancer (13%) thyroid (12%) and liver cancer (11%) were also diagnosed incidentally. In contrast, less than 1% of patients with sarcoma, brain, oesophageal and cervical cancers were diagnosed incidentally.

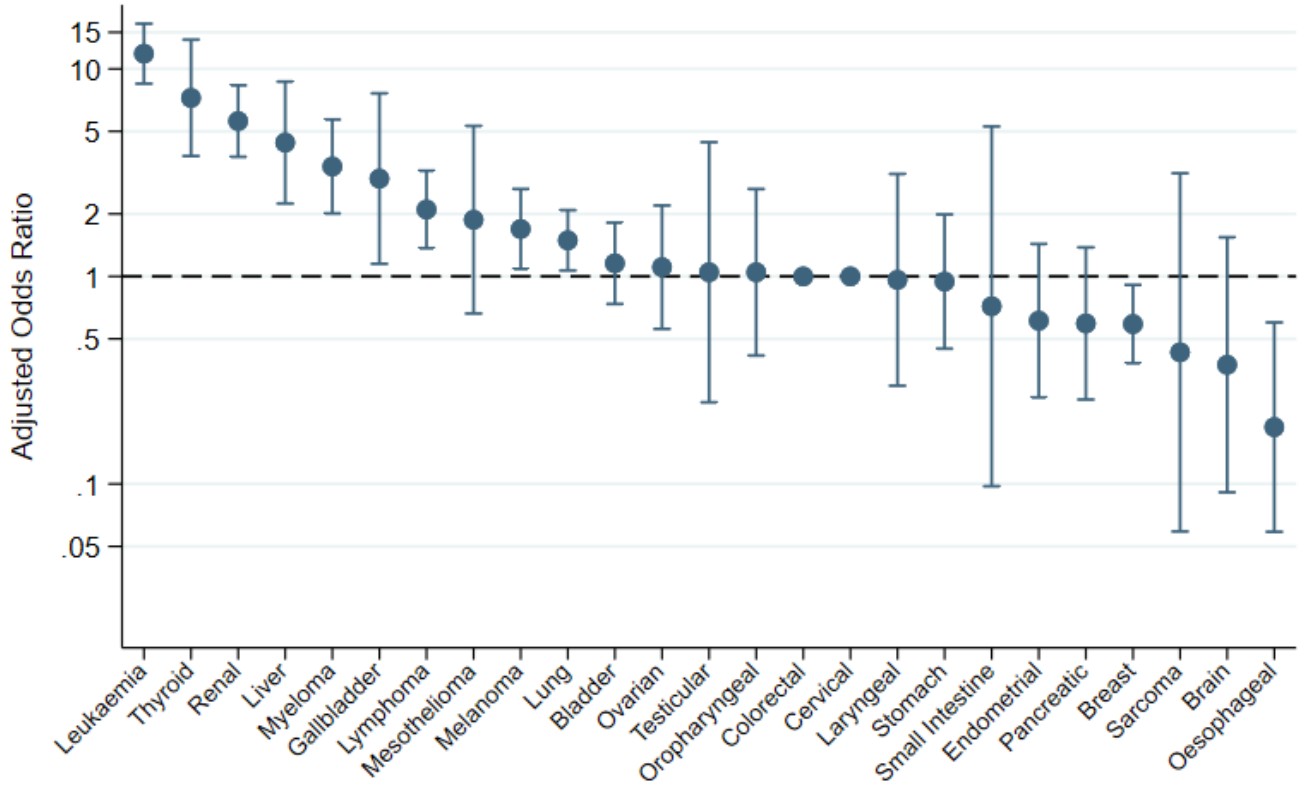

**Figure 1** ORs of incidental versus symptomatic diagnosis of cancer by cancer site (n=13 810; reference group: colorectal cancer). There is no OR for cervical cancer as there were no incidentally diagnosed cases of cervical cancer.

Among the 520 incidentally diagnosed patients, one-fifth (20%, 95% CI: 17% to 23%) were diagnosed with leukaemia, while other common cancer sites included lung (15%, 12% to 18%), colorectal (13%, 10% to 16%) and renal cancers (9%, 7% to 12%) (see figure 2 and online supplementary table 1). There were nine other cancer sites represented among the incidentally diagnosed patients each with 10 or more patients.

Sensitivity analyses (using a broader definition of incidental diagnosis) identified a further 272 cases, increasing the overall estimate of incidental diagnosis to 6% (see online supplementary table 2 and online supplementary figure 2). There was weak evidence to support greater odds of incidental diagnosis among men versus women (adjusted OR (95% CI): 0.84 (0.71 to 1.00)), with

otherwise similar patterns of variation by age group and cancer site as those observed in the main analysis.

### Routes to incidental cancer diagnosis

We identified several clinical scenarios preceding an incidental diagnosis of cancer based on information available for 345 patients (66% of all incidentally diagnosed patients). These are outlined in table 2 and discussed in further detail below.

Many patients received an incidental cancer diagnosis as a result of a clinical encounter for a pre-existing chronic disease in primary or secondary care. This included routine blood or urine testing, as part of chronic disease (or related risk factor) management and monitoring, which revealed abnormalities that led to the diagnosis of

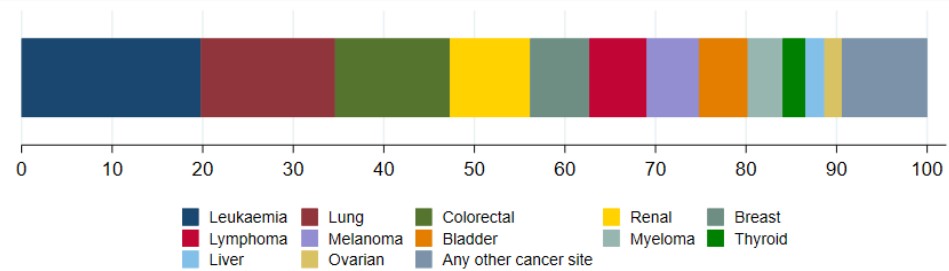

**Figure 2** Commonly diagnosed cancer sites among the incidental cancer patient population; see online supplementary table 1 for frequencies.

**Table 2**  Types of clinical scenarios preceding the incidental diagnosis of cancer

| Clinical scenario | Description and examples |
|---|---|
| Monitoring or managing pre-existing chronic morbidity | Blood or imaging investigations as part of monitoring or management of a chronic morbidity.<br>Eg, haematuria on dipstick urine testing [for diabetes] leading to diagnosis of bladder cancer.<br>Eg, annual blood tests for hypertension leading to diagnosis of leukaemia. |
| Before/after elective surgery for unrelated conditions | Blood or imaging investigations conducted before or after surgery, and more rarely, tumours identified in pathology specimens of tissue resected for other (benign) conditions.<br>Eg, pre-operative chest X-ray leading to diagnosis of lung cancer.<br>Eg, microscopic haematuria noted pre-cataract operation leading to diagnosis of a urological cancer. |
| Staging or follow-up for a previously diagnosed cancer | Blood or imaging investigations carried out as part of staging or follow-up for a previously diagnosed cancer.<br>Eg, scans to ascertain stage at diagnosis of prostate cancer leading to the diagnosis of another urological cancer. |
| Investigation of unrelated acute or new condition/symptoms | Blood or imaging investigations for a new or otherwise acute symptom or condition.<br>Eg, an abdominal ultrasound scan for dyspepsia leading to diagnosis of a urological cancer.<br>Eg, irregular mole noted during health check leading to diagnosis of melanoma. |

otherwise unsuspected cancer. Some patients were diagnosed following blood or imaging investigations before/after elective surgery for unrelated indications, with a small number of patients where tumours were seemingly identified in pathology specimens of tissue resected for non-malignant indications. A small number of patients were diagnosed after blood or imaging investigations carried out as part of staging or follow-up for a previously diagnosed cancer of a different site (eg, scans to ascertain stage at diagnosis of prostate cancer leading to the diagnosis of a renal cancer).

Other cancer patients were diagnosed following the investigation of unrelated acute conditions or presenting symptom(s) unlikely to be related to the subsequent cancer diagnosis. Some of these cases were being investigated for another suspected cancer (eg, a CT scan for a suspected pelvic cancer leading to the diagnosis of colorectal cancer) but for others, the diagnosis was more serendipitous (eg, breast lump found on examination for chest infection).

## DISCUSSION
### Principal findings
Around 1 in 25 patients with cancer in our study population were diagnosed incidentally, with a preponderance of such cases among older patients, and patients with leukaemia, renal cancer, thyroid cancer, liver cancer and multiple myeloma. Clinical scenarios that preceded incidental diagnosis included healthcare encounters relating to previously known chronic conditions, and the investigation of acute or new conditions unrelated to cancer.

### Strengths and limitations
Our study is based on a cohort of cancer patients (diagnosed 2009–2010) and is therefore limited by temporality of the data. However, thus far there have been no subsequent population-based data collections that could enable the detailed examination of the context of presentation in patients subsequently diagnosed with cancer in England. Information on incidental status at diagnosis is not routinely recorded as part of cancer registration data, nor coded as such in administrative databases or patient experience surveys. A strength of our study is that it provides unique evidence about this less well documented diagnostic pathway of cancer, among a large and representative incident cohort characterised by healthcare professionals.

Nevertheless, interpretation of the findings should be mindful of the secondary nature of our analysis. Information on symptoms (or their absence) was based on those recorded in primary care; patients found to be asymptomatic by healthcare professionals participating in the audit may have had symptoms that were not declared or not recorded during the consultation.[19 20] In order to reduce the risk of the resulting bias on analyses, our definition of incidentally diagnosed cancer was deliberately conservative, designed to maximise specificity and reduce the likelihood of patients being mistakenly identified as incidental diagnoses. However, this may have led to the under-estimation of cases; our sensitivity analysis (based on a less conservative definition) indicates that an additional 2% of the study population may have been incidentally diagnosed (online supplementary table 2). Although the true estimates of incidental diagnosis may be higher than those reported, this is unlikely to have biased patterns of variation by cancer site and patient characteristics.

## Comparison with existing literature

Literature examining the prevalence of incidentally diagnosed cancer is limited, although some evidence may be gleaned from studies on incidental findings detected in the context of research studies. Estimates of clinically important incidental findings (including cancer but also other diseases) vary substantially depending on imaging field (whole body or specific organ) and modality however, and participants of research studies are unlikely to be representative of the cancer patient population.[21 22]

Though we were unable to examine potential overdiagnosis, we identified notable proportions of incidentally diagnosed patients with thyroid and renal cancer, and melanoma patients. This is consistent with prior evidence indicating potential overdiagnosis of these cancers.[23–26] A few studies have examined clinical scenarios that result in incidental diagnosis of individual cancer sites such as melanoma, lung cancer and renal cancer.[17 27–29] A study examining self-reported symptoms of haematological cancer patients found that a third of patients did not report any symptoms before diagnosis, with chronic lymphocytic leukaemia patients being particularly prone to being diagnosed incidentally, for example, through blood tests at routine healthcare encounters.[30] Our findings concord with these studies, but additionally show that incidental diagnosis occurs across a range of common and rarer cancers.

## IMPLICATIONS

Currently, there is sparse evidence regarding the prevalence or incidence of incidentally diagnosed cancer, likely due to the challenges in identifying such cases using large administrative healthcare data. Using unique data from an audit initiative, we were able to identify several clinical scenarios resulting in the incidental diagnosis of cancer. This study provides important epidemiological evidence quantifying the frequency of such cases, and characterising the different mechanisms that can lead to an incidental cancer diagnosis.

Our findings indicate that a considerable number of cancer patients are diagnosed with cancer incidentally, without having presented with symptoms related to the subsequent diagnosis. An incidental cancer diagnosis could represent fortuitous early diagnosis of an invasive tumour, and therefore be of clinical benefit for a proportion of patients. However, it could also represent overdiagnosis, which could lead to considerable psychological morbidity and unnecessary treatment.[11]

The frequency of incidental diagnosis, and the relative frequency of the scenarios preceding incidental diagnosis are likely to be affected by system level factors such as approaches to chronic disease monitoring, clinical incentives and thresholds for investigation, availability of imaging services and rates of elective surgery.[31 32] Given increasing levels of multi-morbidity and an ageing population, there is progressively greater use of blood-based testing and imaging studies, which could lead to a greater proportion of patients being diagnosed incidentally, particularly for certain cancer types such as leukaemia.[10] Relatedly, incidental diagnosis of cancer occurred during investigation or follow-up of a pre-existing (unrelated) tumour in a small number of patients. As the survival of patients with cancer continues to improve, this could also become a more frequent route to incidental diagnosis.[33] Further examination of incidentally diagnosed cancer among more contemporary populations, and incidence trends of such diagnoses would be helpful in this regard, particularly given that some of these instances may represent overdiagnosis.

## CONCLUSIONS

In conclusion, we have provided evidence about the frequency and common scenarios leading to incidental diagnosis of cancer. Our findings indicate that this is likely to be occurring in 1 in 25 patients with cancer and calls for further research establishing the prognostic, psychosocial and economic implications of incidentally diagnosed cancer.

**Acknowledgements** We are grateful to all general practitioners and healthcare professionals who were involved in the collection and submission of anonymous data to the audit, and to the respective Cancer Networks, the Royal College of General Practitioners, the former National Cancer Action Team and the former National Clinical Intelligence Network (NCIN) of Public Health England (PHE) for supporting the audit.

**Contributors** MMK, GR and GL conceived the study. MMK conducted all statistical analyses with assistance from GL. MMK wrote the first draft of the manuscript, and prepared the tables and figures, supervised by GL. MMK, GR, SMcP and GL contributed to the interpretation of the results, revised the manuscript and approved the final version of the manuscript.

**Funding** This work was supported by the UK Department of Health as part of the programme of the Policy Research Unit in Cancer Awareness, Screening and Early Diagnosis [grant number no. 106/0001]. The Policy Research Unit in Cancer Awareness, Screening, and Early Diagnosis is a collaboration between researchers from seven institutions (Queen Mary University of London, University College London, King's College London, London School of Hygiene and Tropical Medicine, Hull York Medical School, Durham University and Peninsula Medical School/University of Exeter). GL is supported by Cancer Research UK Clinician Advanced Scientist Fellowship [grant number: C18081/A18180]. GR is Chair, GL Associate Director and MMK Junior Faculty member of the multi-institutional CanTest Collaborative, which is funded by Cancer Research UK (C8640/A23385).The views expressed are those of the authors and not necessarily those of the Department of Health or Cancer Research UK. The funders of the study had no role in the study design, data analysis, data interpretation or writing of the report. The corresponding author had full access to all the data in the study and had final responsibility for the decision to submit for publication.

**Competing interests** None declared.

**Patient consent for publication** Not required.

**Ethics approval** Ethical approval was not required given the anonymous nature of these data.

**Provenance and peer review** Not commissioned; externally peer reviewed.

**Data availability statement** The data used for our analysis can be accessed through the National Cancer Registration and Analysis Service. Enquiries for data access can be made to Public Health England's Office for Data Release (odr@phe.gov.uk).

and indication of whether changes were made. See: https://creativecommons.org/licenses/by/4.0/.

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
