## [Reviewer comments · BMJ Open]

ARTICLE DETAILS

TITLE (PROVISIONAL)	Incidentally diagnosed cancer and commonly preceding clinical scenarios: a cross-sectional descriptive analysis of English audit data
AUTHORS	Koo, Minjoung; Rubin, Greg; McPhail, Sean; Lyratzopoulos, Georgios

VERSION 1 - REVIEW

REVIEWER	Alex Haynes, MD, MPH Massachusetts General Hospital/Harvard Medical School, United States
REVIEW RETURNED	10-Jan-2019

GENERAL COMMENTS	Thank you for the opportunity to review this manuscript. The authors present a cross-sectional analysis of the rate of "incidental" diagnosis of cancer in the UK, using data from a primary care audit. This is an interesting and meaningful area of inquiry and is especially important given the increasing use of advanced imaging techniques. I have a number of queries. 1. The data used here are quite old, nearly a decade. Is there no more contemporaneous data set that may address this issue? I am concerned that changes in the last decade in patterns of clinical care may alter these findings (e.g. management of incidentally identified thyroid nodules, utilization of cross-sectional imaging for a variety of indications).2. The definition of "incidental" and "asymptomatic" appears to be blurred. For example, melanoma is noted as being "incidentally" identified 6% of the time. It is not clear to me what this means. Only in the most advanced cases are patients with melanoma symptomatic. Do these 6% represent patients with metastatic melanoma incidentally noted on imaging? If so, this would negate the voiced concern about overdiagnosis of this condition. If this is in reference to skin lesions noted by clinicians seeing the patients for another reason, then would this be akin to clinical identification of a breast mass? Is the latter considered incidental? Similar concerns arise with other of the diseases included.3. These data are of limited use without outcome data. As the authors correctly state, incidental diagnosis can represent "good luck" or overdiagnosis. The implications of each would be quite different.
---

	Ultimately, this venue of study is worthwhile. The manuscript clearly describes the analysis well, but this study only scratches the surface of the issue. As the authors state, the main conclusion that can be drawn from this study is that there is a significant (albeit small) proportion of patients with cancer who are diagnosed incidentally.
--	---

REVIEWER	Charles Helsper Julius Center for Health Sciences and Primary Care, University Medical Center (UMC) Utrecht, the Netherlands
REVIEW RETURNED	14-Jan-2019

GENERAL COMMENTS	Dear Authors, With great interest, I have read your manuscript “Incidentally diagnosed cancer: population-based evidence on frequency, variation, and commonly preceding clinical scenarios”, which was submitted for publication in BMJ-open. The study described in this manuscript aimed to examine incidentally diagnosed cancer among a cohort of cancer patients diagnosed in England and found that 4% of cancer patients could be defined as incidental diagnoses. This percentage varied between cancer diagnoses and the incidental diagnoses were commonly preceded by a range of clinical scenarios. This study provides some interesting results concerning the incidence rates of incidental diagnoses for several types of cancers. Even though these results are interesting, reading the manuscript left me with essential questions and comments. Before going into detail, I believe these should be first be answered and/or processed first. These questions and comments concern the definition of ‘an incidental diagnosis’, the (relevance of the) analyses performed and the implications of the findings. Definition The definition of ‘an incidental diagnosis’ fully determines the outcomes of the study. This definition however, and e.g. the consequences of this definition on generalizability of findings, remains quite vague. The definition provided is “individuals who were either asymptomatic, or if symptomatic, with presenting symptoms that could not plausibly be related to their subsequent diagnosis. “ The first part of the definition “individuals who were either asymptomatic” raise the question: if there were no symptoms whatsoever, why was the diagnostic follow-up that lead to cancer started? This seems to be partly answered in the results “... included the monitoring or management of pre-existing conditions, routine testing before or after elective surgery, and the investigation of unrelated acute or new conditions”, but a result can hardly be a predefined definition for the main outcome. The second part of the definition that was provided “the absence of symptoms that could not plausibly be related to their subsequent diagnosis”, reminded me of question which is frequently used to illustrate the difficulty of finding cancer in primary care “Which symptom is never a symptom of cancer”. To
---

date, the answer to this question remains as challenging as it is debatable.

This is not solved under “definition and identification of cases” (methods). This section refers to the use of the audit question “what were the MAIN presenting symptoms” (very different from all presenting symptoms) and hereafter refers to an explicit mention of the finding being incidental. How this “ascertainment process” affected the definition is not quite clear and seems debatable.

Since the definition is vital, it needs to be very clear and the consequences of its challenging nature should be discussed more elaborately in the manuscript.

Minor: On this subject, I understand that no (main?) symptom whatsoever may be registered, and particularly for this outcome and the potential clinical implications, a discussion on the difference between ‘not present’ and ‘not registered’ and the impact on findings should be elaborated on in the discussion section.

Data-analysis and relevance

The calculations of incidence seem clear and may be of some relevance (corresponding calculations seem are missing in this section). This data-analysis section however states; “Firstly, we compared the demographic and clinical characteristics of incidentally and non-incidentally diagnosed patients.” This seems to be an irrelevant comparison. In my view, the challenge in clinical practice is not to separate incidentally and non-incidentally diagnosed patients, but to determine which cancer diagnoses are overdiagnosed, and should therefore not be followed up (or actually, who should not be diagnosed in the first place). This brings about the next part which needs to be more clear, which is relevance of findings.

The challenge to find the relevance of findings is illustrated by the last section of the introduction; “Empirical evidence about the frequency and predictors of incidental diagnosis of cancer is needed alongside the consideration of potential overdiagnosis and subsequent clinical, psychological, or economic consequences of this phenomenon.” The frequency is provided by the study, but hereafter the manuscript leaves questions which remain unanswered (as is mentioned in the manuscripts conclusion). These questions bring about a “mental search” for the relevant information throughout the manuscript, which is not satisfied and even fed by more hypotheses (not answers) in the implications section.

Beside this search for the relevant information, some more questions raised by the introduction remain unanswered. E.g.; What does “predictors of incidental diagnosis of cancer” entail? It seems (beside I agree, particularly useful) self-contradictory. And why are they mentioned? This seems to aim for a different goal; finding cancer early that is now missed. This seems to be a different challenge than what this study is about (and since determining the predictors of incidental diagnoses would require

	non-cases, this result beyond the reach of this study and is therefore maybe better left omitted). Also, I agree that information on “..... subsequent clinical, psychological, or economic consequences of this phenomenon” would be useful and seems to be available in the dataset, but this is not included in the study. Why was it not included? The missing information brings about questions concerning relevance and implications. A strong implications section could potentially settle this. But in my view, the implications section does not really seem to describe implications. It mainly presents hypotheses and raises questions, which are not backup up by answers. Clearly stating the clinical relevance of the study (introduction) and findings (implications / conclusions) may help to improve the manuscript.
--	--

VERSION 1 – AUTHOR RESPONSE

Comments from Reviewer 1, Dr Alex Haynes

Thank you for the opportunity to review this manuscript. The authors present a cross-sectional analysis of the rate of "incidental" diagnosis of cancer in the UK, using data from a primary care audit. This is an interesting and meaningful area of inquiry and is especially important given the increasing use of advanced imaging techniques. I have a number of queries.

- We thank Dr Haynes for his positive appraisal of our manuscript.

1. The data used here are quite old, nearly a decade. Is there no more contemporaneous data set that may address this issue? I am concerned that changes in the last decade in patterns of clinical care may alter these findings (e.g. management of incidentally identified thyroid nodules, utilization of cross-sectional imaging for a variety of indications).

- We thank Dr Haynes for highlighting the possible limitations of using data from 2009-10 with regard to possible changes in clinical practice since then (e.g. regarding the availability and frequency of use of investigations). Nonetheless the data source provides a unique opportunity to study our research question. To address the comment, we have added new text to draw attention to the matter in the Strengths and Limitations section which now read as follows (bold indicates updated text):

[Discussion: Strength and Limitations section]

“... Nevertheless, interpretation of the findings should be mindful of the secondary nature of our analysis, and the period of data collection. ...”

Further, in the Implications section of the Discussion, we have added a new sentence (indicated in bold), adding to previous text discussing aspects of the issue of temporality (indicated by underlining):

[Discussion: Implications section]

“... We identified several clinical scenarios that resulted in the incidental diagnosis of cancer; their frequency is likely to be affected by system level factors such as approaches to chronic disease monitoring, incentives and thresholds for investigation, availability of imaging services, and rates of elective surgery [31,32]. Given increasing levels of multi-morbidity and an ageing population, there is

progressively greater use of blood-based testing and imaging studies, which could lead to a greater proportion of patients being diagnosed incidentally, particularly for certain cancer types such as leukaemia [10]. Relatedly, incidental diagnosis of cancer occurred during investigation or follow up of a pre-existing (unrelated) tumour in a small number of patients. As the survival of patients with cancer continues to improve, this could also become a more prevalent route to incidental diagnosis [33]. Further examination of incidentally diagnosed cancer among more contemporary populations would be helpful in this regard.”

2. The definition of "incidental" and "asymptomatic" appears to be blurred. For example, melanoma is noted as being "incidentally" identified 6% of the time. It is not clear to me what this means. Only in the most advanced cases are patients with melanoma symptomatic. Do these 6% represent patients with metastatic melanoma incidentally noted on imaging? If so, this would negate the voiced concern about overdiagnosis of this condition. If this is in reference to skin lesions noted by clinicians seeing the patients for another reason, then would this be akin to clinical identification of a breast mass? Is the latter considered incidental? Similar concerns arise with other of the diseases included.

- We agree that it is important to distinguish between the incidental and asymptomatic diagnosis (see Figure 1, Venn diagram). To address this comment, we have revised the relevant text in the “Definition and identification of cases” section of the Methods which now reads as follows (bold indicates updated text):

“The nature of cancer diagnosis (incidental or non-incidental), was ascertained by examination of the free-text information included in the presenting symptoms data field (answering the audit question “what were the main presenting symptom(s) [of the patient]?”).

Informed by previous literature, we defined the incidental diagnosis of cancer as the diagnosis of cancer in individuals declared as asymptomatic outside the context of population-based screening participation by NACDPC auditors, or individuals noted to have symptoms or clinical signs at presentation that had not been the initial reason for encounter (Davies et al, 2010; Kocher et al, 2016; O’Sullivan et al, 2018a). Cases were initially identified by MMK, and subsequently reviewed and validated by GL and GPR; disagreements were resolved by discussion. Additionally, we identified 520 cases where there was an explicit mention of the incidental nature of diagnosis (e.g. by use of phrases including “accidental finding”; “chance finding”; “incidental”; “opportunistic” or other details regarding circumstances indicating an incidental diagnosis).”

3. These data are of limited use without outcome data. As the authors correctly state, incidental diagnosis can represent "good luck" or overdiagnosis. The implications of each would be quite different.

- We agree that the lack of outcome data represents a limitation of our study and had highlighted this as one of the four bullet points in the “Strengths and Limitations of this study” section.

- The analysis nonetheless contributes important original evidence regarding the frequency of incidental diagnosis and common preceding clinical scenarios/mechanisms, given the paucity of prior literature addressing these questions.

4. Ultimately, this venue of study is worthwhile. The manuscript clearly describes the analysis well, but this study only scratches the surface of the issue. As the authors state, the main conclusion

that can be drawn from this study is that there is a significant (albeit small) proportion of patients with cancer who are diagnosed incidentally.

- We thank Dr Haynes for his positive review and constructive comments and hope that our revisions fully address his queries.

Comments from Reviewer 2, Dr Charles Helsper

With great interest, I have read your manuscript “Incidentally diagnosed cancer: population-based evidence on frequency, variation, and commonly preceding clinical scenarios”, which was submitted for publication in BMJ-open. The study described in this manuscript aimed to examine incidentally diagnosed cancer among a cohort of cancer patients diagnosed in England and found that 4% of cancer patients could be defined as incidental diagnoses. This percentage varied between cancer diagnoses and the incidental diagnoses were commonly preceded by a range of clinical scenarios.

- We thank Dr Helsper for his constructive assessment of our manuscript.

This study provides some interesting results concerning the incidence rates of incidental diagnoses for several types of cancers. Even though these results are interesting, reading the manuscript left me with essential questions and comments. Before going into detail, I believe these should be first be answered and/or processed first.

These questions and comments concern the definition of ‘an incidental diagnosis’, the (relevance of the) analyses performed and the implications of the findings.

Definition

The definition of ‘an incidental diagnosis’ fully determines the outcomes of the study. This definition however, and e.g. the consequences of this definition on generalizability of findings, remains quite vague.

The definition provided is “individuals who were either asymptomatic, or if symptomatic, with presenting symptoms that could not plausibly be related to their subsequent diagnosis. “ The first part of the definition “individuals who were either asymptomatic” raise the question: if there were no symptoms whatsoever, why was the diagnostic follow-up that lead to cancer started? This seems to be partly answered in the results “... included the monitoring or management of pre-existing conditions, routine testing before or after elective surgery, and the investigation of unrelated acute or new conditions”, but a result can hardly be a predefined definition for the main outcome.

The second part of the definition that was provided “the absence of symptoms that could not plausibly be related to their subsequent diagnosis”, reminded me of question which is frequently used to illustrate the difficulty of finding cancer in primary care “Which symptom is never a symptom of cancer”. To date, the answer to this question remains as challenging as it is debatable.

This is not solved under “definition and identification of cases” (methods). This section refers to the use of the audit question “what were the MAIN presenting symptoms” (very different from all

presenting symptoms) and hereafter refers to an explicit mention of the finding being incidental. How this “ascertainment process” affected the definition is not quite clear and seems debatable.

Since the definition is vital, it needs to be very clear and the consequences of its challenging nature should be discussed more elaborately in the manuscript.

- We are grateful to Dr Helsper for his insight. Addressing both reviewers (please see our reply to comment 2 by Reviewer 1 above) we have strengthened the Methods section, which now read as follows (bold indicates updated/modified text):

“The nature of cancer diagnosis (incidental or non-incidental), was ascertained by examination of the free-text information included in the presenting symptoms data field (answering the audit question “what were the main presenting symptom(s) [of the patient]?”).

Informed by previous literature, we defined the incidental diagnosis of cancer as the diagnosis of cancer in individuals declared as asymptomatic outside the context of population-based screening participation by NACDPC auditors, or individuals noted to have symptoms or clinical signs at presentation that had not been the initial reason for encounter (Davies et al, 2010; Kocher et al, 2016; O’Sullivan et al, 2018a). Cases were initially identified by MMK, and subsequently reviewed and validated by GL and GPR; disagreements were resolved by discussion. Additionally, we identified 520 cases where there was an explicit mention of the incidental nature of diagnosis (e.g. by use of phrases including “accidental finding”; “chance finding”; “incidental”; “opportunistic” or other details regarding circumstances indicating an incidental diagnosis).”

Minor: On this subject, I understand that no (main?) symptom whatsoever may be registered, and particularly for this outcome and the potential clinical implications, a discussion on the difference between ‘not present’ and ‘not registered’ and the impact on findings should be elaborated on in the discussion section.

- Many thanks to Dr Helsper for drawing attention to this. We are aware of the possible implications of using primary care data for the recording of symptoms (as opposed to self-reported information from patients after diagnosis) and it is clear that we should have expanded on this further in the manuscript. Therefore, we have added to the “strengths and limitations subheading” of the Discussion, which now read as follows (bold indicates updated text):

“Nevertheless, interpretation of the findings should be mindful of the secondary nature of our analysis, and the period of data collection. Information on symptoms (or their absence) was based on those recorded in primary care; patients found to be asymptomatic by auditors may have had symptoms that were either not declared during the consultation, or else not recorded in their records (Larsen et al, 2014; Leiva et al, 2017). In order to reduce the risk of the resulting bias on analyses, our definition of incidentally diagnosed cancer was deliberately conservative, designed to maximise specificity and reduce the likelihood of patients being mistakenly identified as incidental diagnoses. However, this may have led to the under-estimation of cases; our sensitivity analysis (based on a less conservative definition, see Figure 1) indicates that an additional 2% of the study population may have been incidentally diagnosed (Table S3.1). Although the true estimates of incidental diagnosis may be higher than those reported, this is unlikely to have biased patterns of variation by cancer site and patient characteristics.”

Data-analysis and relevance

The calculations of incidence seem clear and may be of some relevance (corresponding calculations seem to be missing in this section). This data-analysis section however states; "Firstly, we compared the demographic and clinical characteristics of incidentally and non-incidentally diagnosed patients." This seems to be an irrelevant comparison. In my view, the challenge in clinical practice is not to separate incidentally and non-incidentally diagnosed patients, but to determine which cancer diagnoses are overdiagnosed, and should therefore not be followed up (or actually, who should not be diagnosed in the first place). This brings about the next part which needs to be more clear, which is the relevance of findings.

We agree with Dr Helsper that investigating overdiagnosis would be the next logical step for research in this field. In the absence of high-quality data on clinical outcomes however, we are unable to do this as part of the present analysis.

Relatedly, please see our reply to related comment 3 by Reviewer 1, which highlights the empirical contribution of our study with regard to the descriptive epidemiology of incidental diagnosis and related clinical scenarios/mechanisms leading to it.

The challenge to find the relevance of findings is illustrated by the last section of the introduction; "Empirical evidence about the frequency and predictors of incidental diagnosis of cancer is needed alongside the consideration of potential overdiagnosis and subsequent clinical, psychological, or economic consequences of this phenomenon." The frequency is provided by the study, but hereafter the manuscript leaves questions which remain unanswered (as is mentioned in the manuscript's conclusion). These questions bring about a "mental search" for the relevant information throughout the manuscript, which is not satisfied and even fed by more hypotheses (not answers) in the implications section.

Beside this search for the relevant information, some more questions raised by the introduction remain unanswered. E.g.; What does "predictors of incidental diagnosis of cancer" entail? It seems (beside I agree, particularly useful) self-contradictory. And why are they mentioned? This seems to aim for a different goal; finding cancer early that is now missed. This seems to be a different challenge than what this study is about (and since determining the predictors of incidental diagnoses would require non-cases, this result is beyond the reach of this study and is therefore maybe better left omitted).

- We thank Dr Helsper for identifying areas where we could improve the communication and contextualisation of our research aims. Given the aforementioned paucity in research in this area, we feel it is appropriate to compare incidental versus non-incidental cancer cases in order to fully describe the characteristics of this often overlooked patient population and to elucidate possible clinical scenarios and mechanisms leading to such diagnoses.

We have revised the Introduction section, which now reads as follows (bold indicates updated text):

"... Since incidental diagnoses are characterised by the absence of tumour related symptoms, it is plausible that some patients with incidentally detected cancer could be overdiagnosed, whereby the detected cancer would not have otherwise caused symptoms in the patient's lifetime (Esserman et al, 2014). Concerns about overdiagnosis thus far have largely focused on screening-detected cancers (e.g. breast cancer), but it may be also occurring in other contexts (Jenniskens et al, 2017; Davies et al, 2018). Ahead of considering the clinical, psychological, or economic consequences associated with incidental diagnosis (including the potential for overdiagnosis), we need to address gaps in our knowledge about the frequency and characteristics of incidentally diagnosed cancer.

We therefore aimed to examine the frequency of incidental diagnosis among an incident cohort of cancer patients; compare the characteristics of incidentally vs non-incidentally diagnosed patients; and examine common pathways and mechanisms likely to lead to incidental diagnosis of cancer.”

Also, I agree that information on “..... subsequent clinical, psychological, or economic consequences of this phenomenon” would be useful and seems to be available in the dataset, but this is not included in the study. Why was it not included?

- Unfortunately we do not have high-quality information on staging, nor information on psychological outcomes or other patient reported outcome measures (and the costs associated with each route of diagnosis that would enable cost-effectiveness/cost-benefit analyses). The NACDPC dataset contains anonymised information which cannot be linked to other datasets to enable this type of analysis. See our revisions to the Introduction above.

The missing information brings about questions concerning relevance and implications. A strong implications section could potentially settle this. But in my view, the implications section does not really seem to describe implications. It mainly presents hypotheses and raises questions, which are not backup up by answers. Clearly stating the clinical relevance of the study (introduction) and findings (implications / conclusions) may help to improve the manuscript.

- We thank Dr Helsper for his insightful comments. These have led to revisions of the Discussion (Conclusion) sections, which now read as follows (bold indicates updated text):

“In conclusion, we have provided evidence about the frequency and common scenarios leading to incidental diagnosis of cancer. Our findings indicate that this is likely to affect around one in 25 cancer patients and calls for further research establishing the prognostic, psychosocial and economic implications of incidentally diagnosed cancer.”

Please also see our reply to comment 3 by Reviewer 1, which highlight the empirical contribution of our study.

VERSION 2 – REVIEW

REVIEWER	Alex Haynes Massachusetts General Hospital/Harvard Medical School United States
REVIEW RETURNED	09-Apr-2019

GENERAL COMMENTS	Thank you for the opportunity to review this revised manuscript. While the authors have addressed some of the framing questions and clarified methods, they have not fully answered the question about the temporality of the data. Do these data not exist for a more recent era? The value of publishing an analysis based on decade old data is limited and would only be recommended if there is not a more recent data set. Ultimately, the reader is interested in what the epidemiology of incidentally identified
---

	cancer is now (or as close an approximation to “now” as possible) and the authors should use the best available data to do this. If this truly is the most recent data, then please state so. Otherwise, completely understanding the inconvenience, the authors should obtain more recent data and re-run the analysis to avoid the findings being stale from the day they are published. A minor point for table 3: it might make sense to list myeloma adjacent to leukaemia and lymphoma so that the hematologic malignancies are all together. Another minor point: Consider dividing leukaemia into acute and chronic leukaemias. I suspect that CLL represents a large proportion of these patients, but it may be helpful to know.
--	---

REVIEWER	Charles Helsper Julius Center for Health Sciences and Primary Care, UMC Utrecht / Utrecht University, the Netherlands
REVIEW RETURNED	09-May-2019

GENERAL COMMENTS	Dear Authors, Thank you for the revisions on your manuscript “Incidentally diagnosed cancer: population-based evidence on frequency, variation, and commonly preceding clinical scenarios”. I believe the adaptations have improved the paper. Even though the paper has improved, I believe some additional clarifications could further improve the interpretability of the findings. The definition incidental/asymptomatic diagnosis has improved, since it is now clearer WHAT has been done. New text provided by authors: “The nature of cancer diagnosis (incidental or non-incidental), was ascertained by examination of the free-text information included in the presenting symptoms data field (answering the audit question “what were the main presenting symptom(s) [of the patient]?”). Informed by previous literature, we defined the incidental diagnosis of cancer as the diagnosis of cancer in individuals declared as asymptomatic outside the context of population-based screening participation by NACDPC auditors, or individuals noted to have symptoms or clinical signs at presentation that had not been the initial reason for encounter (Davies et al, 2010; Kocher et al, 2016; O’Sullivan et al, 2018a). Cases were initially identified by MMK, and subsequently reviewed and validated by GL and GPR; disagreements were resolved by discussion. Additionally, we identified 520 cases where there was an explicit mention of the incidental nature of diagnosis (e.g. by use of phrases including “accidental finding”; “chance finding”; “incidental”; “opportunistic” or other details regarding circumstances indicating an incidental diagnosis).” This section does however deserves a short addition on HOW this has been done.
--

	How was the distinction between incidental and non-incidental made? Which definitions were used? Was this based on literature or clinical expertise? The sentence addressing this does not make exactly clear how this essential process of selection was performed. “Informed by previous literature, we defined the incidental diagnosis of cancer as the diagnosis of cancer in individuals declared as asymptomatic outside the context of population-based screening participation by NACDPC auditors, or individuals noted to have symptoms or clinical signs at presentation that had not been the initial reason for encounter” The description of the limitations of the definition of asymptomatic has much improved (discussion). Given the relevance of this definition: could you consider changing “asymptomatic”, to “diagnoses without registered symptoms”? Either in the methods section / Figure 1 (no symptoms = no registered symptoms), or perhaps even throughout the manuscript. As for the comparison between demographic and clinical characteristics of incidental versus non-incidental diagnosis, given the very different clinical situations, I am still not convinced of the usefulness for daily practice. I still believe this outcome is overemphasized in the manuscript (provided OR). I do however understand that this is a matter of taste, and on a policy level, I can understand the potential relevance. Perhaps it would be helpful to make this distinction (clinical versus policy level) and the corresponding implications more clear in the discussion section. Minor:  - The calculations of incidence (main outcome) remain missing under Data analysis. A short mentioning of the numerator (incidental diagnosis?) and denominator (and how “studied cancer population” was determined) seems befitting. - Given the datacollection period. Please add a short comment on potential changes in the health care system (or other) that could have influenced findings. E.g. has there been a change in screening programmes? - Some limitations of the dataset were described, which have hampered suggested addition to the manuscript (e.g.: inability to link -> no: consequences of overdiagnosis / staging) Perhaps the authors could add these limitations of the dataset to the discussion, to describe why some relevant additional analyses were not possible.
--	--

VERSION 2 – AUTHOR RESPONSE

Comments from Reviewer 1, Dr Alex Haynes

Thank you for the opportunity to review this revised manuscript. While the authors have addressed some of the framing questions and clarified methods, they have not fully answered the question about the temporality of the data. Do these data not exist for a more recent era? The value of publishing an analysis based on decade old data is limited and would only be recommended if there is not a more recent data set. Ultimately, the reader is interested in what the epidemiology of incidentally identified cancer is now (or as close an approximation to “now” as possible) and the authors should use the best available data to do this. If this truly is the most recent data, then please state so. Otherwise,

completely understanding the inconvenience, the authors should obtain more recent data and re-run the analysis to avoid the findings being stale from the day they are published.

- We thank Dr Haynes for the supportive comments regarding improvements in the manuscript. We agree that more timely data would be more useful, but there has been no prior or subsequent comparable data source to support a study of incidentally diagnosed cancer in England. The NACDPC remains the single national audit initiative in England that collected detailed free-text data from healthcare professionals (usually a primary care physician) characterising the type and context of symptomatic presentations leading to diagnosis of cancer. Future possible repeats of this study would require major logistical undertakings and are not in scope of our project. Nonetheless, should there be a repeat study, ours will be useful for benchmarking comparisons.

- Given the comment, we have clarified this matter further throughout the manuscript (bold indicates updated text):

[Introduction]

We therefore aimed to examine the frequency of incidental diagnosis among an incident cohort of cancer patients; compare the characteristics of incidentally vs non-incidentally diagnosed patients; and examine common pathways and mechanisms likely to lead to incidental diagnosis of cancer using a unique data source relating to a national quality improvement initiative in England.

[Methods: Study design and population section]

“We analysed cross-sectional data collected as part of the English National Audit of Cancer Diagnosis in Primary Care (NACDPC) [14]. Briefly, health professionals from 1,170 participating general practices (representing 14% of practices in England) provided information on the diagnostic pathway for a consecutive sample of patients diagnosed with cancer during April 2009–2010. Participating practices were comparable to non-participating practices in (former) respective Cancer Networks, and the patient population was broadly representative of the contemporary national incident cancer patient cohort [14,15]. Unique to this audit, clinicians participating in the NACDPC provided extensive information regarding the main presenting symptoms, cancer diagnosis, demographic characteristics, and route of diagnosis for each patient based on primary care records.”

[Discussion: Strength and Limitations section]

“Our study is based on a cohort of cancer patients (diagnosed 2009–10) and is therefore limited by temporality of the data. However, thus far there have been no subsequent population-based data collections that could enable the detailed examination of the context of presentation in patients subsequently diagnosed with cancer in England. Information on incidental status at diagnosis is not routinely recorded as part of cancer registration data, nor coded as such in administrative databases or patient experience surveys. A strength of our study is that it provides unique evidence about this less well documented diagnostic pathway of cancer, among a large and representative incident cohort characterised by healthcare professionals.”

A minor point for table 3: it might make sense to list myeloma adjacent to leukaemia and lymphoma so that the hematologic malignancies are all together.

- There was no Table 3, so we interpret Dr Haynes’s comment as likely referring to Table 1 (Characteristics of incidental cancer patients versus non-incidentally cancer patients, and crude/adjusted odds ratios of incidental status (n=13,810)), where cancer site is presented by proportion of patients diagnosed incidentally (descending order). Having considered the suggestion, we believe using the current format is optimal for ease of reading and data assimilation by the reader.

Another minor point: Consider dividing leukaemia into acute and chronic leukaemias. I suspect that CLL represents a large proportion of these patients, but it may be helpful to know.

- We agree that dividing leukaemia into acute and chronic subtypes would have been ideal but unfortunately the NACDPC did not collect data at that level of granularity for this site (see also Lyratzopoulos et al., British Journal of Cancer 2013, <https://dx.doi.org/10.1038/bjc.2013.1>).

- We have added text to the Methods and a footnote to Table 1 to make this clear:

[Methods, Definition and identification of cases]

- "... Information was available on the patient's sex and age group, and cancer site (categorised as Bladder, Brain, Cervical, Colorectal, Endometrial, Gallbladder, Leukaemia (of any type), Laryngeal, Liver, Lung, Lymphoma, Melanoma, Mesothelioma, Multiple Myeloma, Oesophageal, Oropharyngeal, Ovarian, Pancreatic, Renal, Sarcoma (of any type), Small Intestine, Stomach, Testicular, Thyroid and Vulval) [14]. ..."

[Results, Table 1]

"*No information was available on leukaemia or sarcoma type." [new footnote]

Comments from Reviewer 2, Dr Charles Helsper

Thank you for the revisions on your manuscript "Incidentally diagnosed cancer: population-based evidence on frequency, variation, and commonly preceding clinical scenarios".

I believe the adaptations have improved the paper.

Even though the paper has improved, I believe some additional clarifications could further improve the interpretability of the findings.

The definition incidental/asymptomatic diagnosis has improved, since it is now clearer WHAT has been done. New text provided by authors:

"The nature of cancer diagnosis (incidental or non-incidental), was ascertained by examination of the free-text information included in the presenting symptoms data field (answering the audit question "what were the main presenting symptom(s) [of the patient]?").

Informed by previous literature, we defined the incidental diagnosis of cancer as the diagnosis of cancer in individuals declared as asymptomatic outside the context of population-based screening participation by NACDPC auditors, or individuals noted to have symptoms or clinical signs at presentation that had not been the initial reason for encounter (Davies et al, 2010; Kocher et al, 2016; O'Sullivan et al, 2018a). Cases were initially identified by MMK, and subsequently reviewed and validated by GL and GPR; disagreements were resolved by discussion. Additionally, we identified 520 cases where there was an explicit mention of the incidental nature of diagnosis (e.g. by use of

phrases including “accidental finding”; “chance finding”; “incidental”; “opportunistic” or other details regarding circumstances indicating an incidental diagnosis).”

This section does however deserves a short addition on HOW this has been done.

How was the distinction between incidental and non-incidental made? Which definitions were used? Was this based on literature or clinical expertise?

The sentence addressing this does not make exactly clear how this essential process of selection was performed.

“Informed by previous literature, we defined the incidental diagnosis of cancer as the diagnosis of cancer in individuals declared as asymptomatic outside the context of population-based screening participation by NACDPC auditors, or individuals noted to have symptoms or clinical signs at presentation that had not been the initial reason for encounter”

- We thank Dr Helsper for his supportive comments and for encouraging further clarification of how incidentally diagnosed patients were identified. We identified incidentally diagnosed cancer based on a) explicit mention of the incidental nature of diagnosis or b) other details on the context / clinical scenarios clearly indicating an incidental diagnosis.

- The former group (a), relies on clinical expertise of the original auditors when completing the NACDPC (usually attending primary care physicians). The latter group (b), relies on judgement by the study authors based on the information included in the audit data, the clinical knowledge of the authors (YL and GPR), and the available literature.

- Given the comment, we have improved the manuscript in two parts of the Methods section (Definition and identification of cases; and Sensitivity analysis) which now read as follows (bold indicates updated text):

[Methods – Definition and identification of cases]

“The nature of cancer diagnosis (incidental or non-incidental), was ascertained by examination of the free-text information included in the presenting symptoms data field (answering the audit question “what were the main presenting symptom(s) [of the patient]?”).

Tumours were deemed to have been diagnosed incidentally if the incidental nature of diagnosis was explicitly recorded by the participating healthcare professional (indicated by phrases including “accidental finding”; “chance finding”; “incidental”; “opportunistic”), or if the clinical circumstances described were consistent with incidental identification based on clinical knowledge (GL and GPR) and prior literature [5,6,17]. Cases were initially identified by MMK, and subsequently reviewed and validated by GL and GPR; disagreements were resolved by discussion. ...”

[Methods – Sensitivity analysis]

“We performed sensitivity analyses expanding the definition of incidental diagnosis of cancer to include an additional 272 patients without registered presenting symptoms, and/or with abnormal clinical findings (without recorded symptoms) to the audit question “what were the main presenting symptom [of the patient]?” . ”

Additionally, we have removed Figure 1 as it is superfluous, and potentially confusing.

The description of the limitations of the definition of asymptomatic has much improved (discussion). Given the relevance of this definition: could you consider changing “asymptomatic”, to “diagnoses without registered symptoms”? Either in the methods section / Figure 1 (no symptoms = no registered symptoms), or perhaps even throughout the manuscript.

- We thank Dr Helsper for his positive appraisal of the edits to the Discussion section and have changed mentions of “asymptomatic” to “no registered symptoms” throughout the manuscript where applicable (noting that we have now removed Figure 1, see comment above).

As for the comparison between demographic and clinical characteristics of incidental versus non-incidental diagnosis, given the very different clinical situations, I am still not convinced of the usefulness for daily practice. I still believe this outcome is overemphasized in the manuscript (provided OR). I do however understand that this is a matter of taste, and on a policy level, I can understand the potential relevance. Perhaps it would be helpful to make this distinction (clinical versus policy level) and the corresponding implications more clear in the discussion section.

- We agree with Dr Helsper in that our findings are unlikely to have direct relevance to clinical practice. Rather, our paper should be seen as a ‘proof of concept’ study that provides basic epidemiological evidence elucidating the subject of incidental diagnosis of cancer. We believe this contributes meaningfully to existing literature as current evidence is largely anecdotal, with only a few studies focusing on imaging (see cited references). In order to make this clearer, we have edited the Discussion section to now read as follows (bold indicates updated text):

[Discussion: Implications section]

“Currently, there is sparse evidence regarding the prevalence or incidence of incidentally diagnosed cancer, likely due to the challenges in identifying such cases using large administrative healthcare data. Using unique data from an audit initiative, we were able to identify several clinical scenarios resulting in incidental diagnosis of cancer. This study provides important epidemiological evidence quantifying the frequency of such cases, and characterising the different mechanisms that can lead to an incidental cancer diagnosis.”

Minor:

- The calculations of incidence (main outcome) remain missing under Data analysis. A short mentioning of the numerator (incidental diagnosis?) and denominator (and how “studied cancer population” was determined) seems befitting.

- We present the frequency (prevalence) of incidentally diagnosed cancer (rather than incidence). Derivation of the study population for the main analysis is described in Supplementary figure 1, which shows the derivation of the numerator and denominator for calculating the frequency of incidental diagnosis.

- Given the data collection period. Please add a short comment on potential changes in the health care system (or other) that could have influenced findings. E.g. has there been a change in screening programmes?

- While we acknowledge there may have been changes to the three cancer screening programmes in the UK, this is unlikely to influence our findings given that we exclude screen-detected cancer patients. Nevertheless, we agree with Dr Helsper (and indeed Dr Haynes, see first comment) on the limitations of the study given that data collection occurred in 2009-10 and have added to the Strengths and Limitations section of the Discussion, which now read as follows (bold indicates updated text):

- [Discussion: Strength and Limitations section]

- “Our study is based on a cohort of cancer patients (diagnosed 2009–10) and is therefore limited by temporality of the data. However, thus far there have been no subsequent population-based data sources that could enable the detailed examination of the context of presentation among patients subsequently diagnosed with cancer in England. Information on incidental status at diagnosis is not routinely recorded as part of cancer registration data, nor coded as such in administrative databases or patient experience surveys. A strength of our study is that it provides unique evidence about this less well documented diagnostic pathway of cancer, among a large and representative incident cohort characterised by healthcare professionals.”

- Furthermore, we discuss the likely influence of system level factors that could influence the frequency of incidental diagnosis, and the need for further examination of this phenomenon among more recent populations in the Implications section, see the new text above (page 4).

- Some limitations of the dataset were described, which have hampered suggested addition to the manuscript (e.g.: inability to link -> no: consequences of overdiagnosis / staging)

Perhaps the authors could add these limitations of the dataset to the discussion, to describe why some relevant additional analyses were not possible.

- We agree that it is unfortunate that we could not examine clinical outcomes such as staging – indeed this is why we aimed to describe the phenomenon of incidentally diagnosed cancer rather than examining the possibility of overdiagnosis. We believe that the strengths of this dataset, namely the ability to ascertain incidentally diagnosed cancer cases and describe possible scenarios preceding diagnosis, are considerable in the absence of other means to examine this phenomenon – please see also our reply to an earlier comment, which also addresses this comment.

VERSION 3 - REVIEW

REVIEWER	Charles Helsper Julius Center for health sciences and primary care, UMC Utrecht / Utrecht university
REVIEW RETURNED	01-Jul-2019

GENERAL COMMENTS	Dear Authors, Thank you for the responses to the comments and the changes made to the manuscript. In my view, you have adequately processed my comments and I look forward to your future work. Best regards, Charles
--